# Mechanical Properties and Fracture Behavior of Laser Powder-Bed-Fused GH3536 Superalloy

Haohan Ni [1,†], Qi Zeng [2,3,†], Kai Zhang [4], Yingbin Chen [1] and Jiangwei Wang [1,5,*]

1   Center of Electron Microscopy, State Key Laboratory of Silicon Materials, School of Materials Science and Engineering, Zhejiang University, Hangzhou 310027, China; haohan_ni@zju.edu.cn (H.N.); yingbin_chen@zju.edu.cn (Y.C.)
2   School of Aeronautics and Astronautics, Zhejiang University, Hangzhou 310027, China; 12024082@zju.edu.cn
3   AECC Hunan Aviation Plant Research Institute, Zhuzhou 412002, China
4   School of Materials and Chemistry, University of Shanghai for Science and Technology, Shanghai 200093, China; kai.zhang@monash.edu
5   Wenzhou Key Laboratory of Novel Optoelectronic and Nano Materials, Institute of Wenzhou, Zhejiang University, Wenzhou 325006, China
*   Correspondence: jiangwei_wang@zju.edu.cn
†   These authors contributed equally to this work.

**Abstract:** Heat treatment (HT) is an important approach to tune the structure and mechanical properties of as-printed or hot-isostatic-pressed (HIPed) additive manufacturing materials. Due to the carbide precipitates extensively existing after HT with air cooling, this paper studies the microstructure and mechanical behavior of laser powder-bed-fused (L-PBFed) GH3536 superalloy with laminar carbide precipitates at grain boundaries. By comparing with air-cooling samples and water-quenched samples, the results revealed that air cooling often introduced precipitates at grain boundaries, which impede the plastic deformation and are prone to lead to severe transgranular cracks on the fracture surface, contributing to a higher strain-hardening rate but lower ductility of HTed sample. Water quench can largely eliminate the grain-boundary precipitates, contributing to an optimized ductility even with smaller grain size. This work provides more details on the precipitate-deformation relation after HT.

**Keywords:** additive manufacturing; heat treatment; nickel-based superalloy; precipitates

## 1. Introduction

Nickel-based superalloys are the most widely used metallic materials as hot parts of gas turbine [1,2]. The GH3536 superalloy (Hastalloy X) is a typical solid-solution-strengthened nickel-based superalloy with high oxidation and corrosion resistances as well as superior mechanical properties at high temperature [3]. However, conventional manufacturing methods such as forging and casting, which were used to build hot parts, are cumbersome in steps and time-/material-consuming [4]. Meanwhile, melt defects, including carbides and segregations [5,6], hinder the improvement of the high-temperature performance of as-built GH3536 superalloys. Since 3D-printing technology has shown great potential in replacing or optimizing conventionally industrial manufacturing technologies, applying laser-additive manufacturing in building superalloy has been also wildly investigated [7].

Laser powder-bed fusion (L-PBF) is one of the laser-based additive manufacturing methods with the capability of higher shape precision, lower material waste, and fewer processing steps than conventional manufacturing technologies [8,9]. The mechanical properties of as-printed samples are often reported as higher-strength but lower-ductility compared to the casting or wrought parts [10,11]. Although a good combination of strength and ductility can be achieved in some metals and alloys, the spatial heating and cooling

cycle during L-PBF often induce nonequilibrium and anisotropic structures [12,13], as well as high-density dislocations, which greatly impair the mechanical properties of as-printed materials. For example, the L-PBF-induced molten pool boundaries, crystallographic texture, micropores, and cracks can lead to low ductility and mechanical anisotropy of L-PBFed materials [13–16]. To eliminate these drawbacks, numerous efforts have been devoted to fabricating materials with excellent properties. Generally, by optimizing the parameters of the L-PBF process, including laser power [17], scan speed [18,19], and scan strategy [20,21], an enhanced relative density and weakened texture can be obtained. However, some metallurgical defects such as solidification cracking [22], liquation cracking [23], and pores [17] are difficult to eliminate via the optimization of printing parameters. Therefore, post-treatments, including heat treatment (HT) and hot isostatic pressing (HIP) are often introduced to reduce metallurgical defects of additive manufacturing for an optimized mechanical property.

After L-PBF fabrications, an anisotropic structure with high density of dislocations and metallurgical defects is formed because of the effects of thermal gradients, heat cycling, and solidification conditions. Fine structures with numerous dislocations lead to high tensile strength, but the anisotropy and metallurgical defects often cause limited ductility of L-PBFed Hastalloy X [24–26]. Since hot parts are sensitive to pores and cracks due to stress concentration especially under fatigue loading [27,28], HIP is often applied to homogenize the microstructure and reduce the metallurgical defects of L-PBFed metallic parts. However, carbide precipitates in grain interiors and grain boundaries after HIP have been extensively reported in previous studies, and these precipitated particles at grain boundaries often result in lower mechanical properties and thermal stability of L-PBFed Hastalloy X [29,30]. Thus, HIP is commonly combined with HT to simultaneously eliminate pores and cracks and dissolve the precipitates introduced by HIP. Noting that HT could also introduce precipitates [31] and that the porosity of the HIPed sample is prone to regrow after HT [32,33], it is necessary to reconsider the role of HT and to optimize HT parameters to achieve excellent mechanical properties.

Generally, during HT at solid-solution temperature, the residual stress, strong texture, submicrometric cell structures, and part of metallurgical defects can be greatly eliminated, contributing to an enhanced elongation and relative density of L-PBFed Hastalloy X [29,34]. However, the grain size and the density of precipitates are strongly correlated with HT parameters. Conventionally, the HT of the solid-solution alloy is mainly affected by homogenizing time and cooling conditions. Previous studies [29,30] tended to conduct a sufficient homogenizing time (e.g., 2 h) with air cooling to make full recrystallization and solid solution of Hastalloy X when merely applying HT. However, in the case of HIP combining with HT, a shorter homogenizing time with water quench was applied in the HIPed sample [35]. Hence, the mechanical properties and fracture behavior of Hastalloy X after different HT processes need further investigations in order to establish a full structure–property relation.

In this work, we investigate the mechanical behavior of L-PBFed and HTed GH3536 superalloy. Mechanical properties were measured at room temperature, and the influences of carbide precipitates at grain boundaries were discussed by comparing with samples under different HT conditions. The deformation mechanisms of as-printed and HTed samples were investigated. This paper reports the HT optimization for improving ductility of HTed metal parts and provides more details on the precipitate-deformation relation after HT.

## 2. Materials and Methods

GH3536 powders (sizes ranged from 10–53 μm, ordered from Avimetal Powder Metallurgy Technology Co., Ltd., Beijing, China) were used in this study. The nominal composition of as-received powders and the compositions of as-printed sample confirmed by energy-dispersive spectroscopy (EDS) quantification at beam energy of 20 kV are shown in Table 1. Because the elements with atomic number of less than 11 are not reliable in EDS, carbon and boron were not included. At least 10 points of EDS scanning were acquired and the molten-pool boundaries were avoided. L-PBF fabrication was carried out in EOS

M290. The parameter optimization was carried out prior to this study by using Doelhert parameter-optimization approach (with the details found in [36]), with the initial parameter ranges as follows: laser power 195–360 W, laser scan speed 930–1200 mm/s, and hatch distance 0.08–0.13 mm. The layer thickness was fixed as 40 μm. Four rounds of parameter optimization with 60 samples were applied. During the L-PBF process, the printing parameters were set as laser power 245 W, scan speed 1198.6 mm/s, hatch distance 0.08 mm, and layer thickness 0.04 mm. A substrate preheating of 80 °C was used for the L-PBF process to reduce the residual stress generated during the fabrication process. After fabrication, the relative density of the as-fabricated sample was measured to be 99.8% based on the image-characterization method described in the previous study [37].

**Table 1.** Nominal and measured (EDS quantification) compositions of L-PBFed GH3536 superalloy.

| Element | Ni | Cr | Fe | Mo | Co | Mn | W | Si | C | B |
|---|---|---|---|---|---|---|---|---|---|---|
| Nominal wt.% | Bal. | 22.0 | 18.0 | 9.0 | 1.5 | Max 1.0 | 0.6 | Max 1.0 | 0.1 | Max 0.008 |
| Measured wt.% | Bal. | $21.93 \pm 0.17$ | $18.83 \pm 0.16$ | $9.95 \pm 0.45$ | $1.57 \pm 0.04$ | $0.13 \pm 0.12$ | $0.84 \pm 0.14$ | $0.06 \pm 0.04$ | — | — |

Subsequently, we tried the different times of HT with air cooling and water quenching, including 0.5/1.0/1.5/2.0 h. We found that two hours' HT followed by air cooling and one hour followed by water quenching enabled the samples' best mechanical properties and similar grain size. Thus, these two samples were used in this paper: (1) homogenized and recrystallized at 1177 °C for 2 h followed by air cooling (1177-AC) in an indoor environment; (2) annealed at 1177 °C for 1 h followed by water quenching (1177-WQ). All the samples were heated with the furnace at a rate of 10 °C/min and without protection from atmosphere. After heat treatments, samples for microstructural characterizations and uniaxial tensile testing were cut perpendicular to the building direction of the L-PBF samples. The average grain size was determined by a linear intercept method according to the ASTM E112 standard (excluding the annealing twin boundaries (TBs)), and three images were counted in every HTed sample. For tensile testing, dog-bone samples with the gauge sections of $2.0 \times 2.0 \times 9.5$ mm$^3$ were machined perpendicular to building direction by electrodischarge, followed by mechanical gridding with SiC paper. Room-temperature tensile tests were conducted in a screw-driven SUST-CMT5000GL mechanical testing machine at a strain rate of $10^{-3}$ s$^{-1}$.

For microstructural characterizations, samples for optical microscope observation were firstly cut horizontal to building direction and carefully ground by 200, 400, 800, 1200, and 2000 grit SiC paper in turn. Then two-step mechanical polishing by diamond-particle suspension with a particle size of 5 μm and 1 μm was conducted to make the surface of the sample as bright as a mirror. Samples for scanning electron microscope observation were further electro-etched in an Oxalic-acid saturated solution with a voltage potential of 6 V for 15–20 s at room temperature [24]. Hitachi SU-70 field-emission scanning electron microscope (SEM) equipped (Hitachi, Tokyo, Japan) with an Oxford energy-dispersive X-ray spectroscopy (EDS) detector (Oxford Instruments, Oxfordshire, United Kingdom) was employed to analyze the microstructure and element distribution. Transmission electron microscopy (TEM) samples were punched as discs with a diameter of 3 mm, and then twinjet-electropolished in a methanol solution containing 5 vol.% perchloric acid at −20 °C. Scanning TEM (STEM) images were carried out using a spherical aberration-corrected FEI Titan G2 80–200 ChemiSTEM (Thermo Fisher Scientific, Hillsboro, OR, United States) operating at 200 kV equipped with a high-angle annular dark-field (HAADF) detector and a bright-field (BF) detector. X-ray diffraction (XRD) was performed on a copper target ($\lambda$ = 1.5406 Å) PANalytical X'Pert PRO system (PANalytical B.V., Almelo, The Netherlands) with generator voltage 40 kV and tube current 40 mA to determine the crystal structure of GH3536 superalloy. The scan-step size was 0.02 degrees.

## 3. Results

### 3.1. Microstructures

With the process of parameter optimization using the Doehlert experimental design approach, a dense GH3536 superalloy was fabricated by L-PBF. Figure 1 shows the high densification of the as-fabricated part horizontal to the building direction with no visible crack or print defect. Figure 2 shows the representative microstructures of the sample perpendicular to the building direction. As shown in Figure 2a, the as-fabricated samples contain numerous overlapped melting pools. These melting pools consist of multiple grains defined by the orientation of dendrites (Figure 2b). Bright-field STEM characterization shows that each grain in the melting pool contains many polygonal dislocation cells, with an average diameter of ~0.6 μm (Figure 2c). Figure 2d shows that the boundaries of these dislocation cells are composed of numerous tangled dislocations, with some residual precipitated particles distributed randomly in the interior and at the boundary of the dislocation cell. EDS mapping of this region further demonstrates that the precipitated particles mainly consist of Cr and Mo, while the segregated elements at cell boundaries contain Fe, Cr, Mo, and Mn (Figure 2e). According to previous studies [38,39], dislocation structures and precipitates are formed due to the rapid heating and cooling during printing, where the density of dislocations increases with thermal distortion and is affected by geometric constraints.

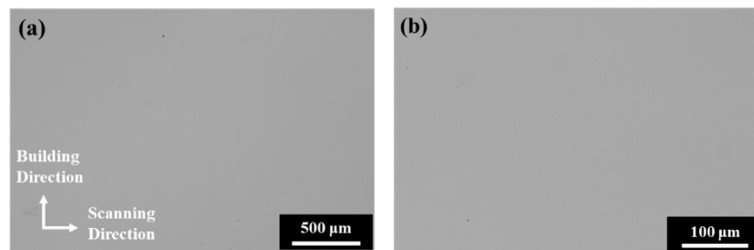

**Figure 1.** Optical micrographs of the as-fabricated GH3536 superalloy horizontal to building direction. (**a**) in low magnification, (**b**) in middle magnification.

Subsequently, the as-fabricated GH3536 superalloy was heat-treated under different conditions. Figure 3a,b show the microstructures of HTed samples, including 1177-AC (Figure 3a) and 1177-WQ (Figure 3b). After HT, the large melting-pool boundaries and dendrites were largely dissolved and the whole sample developed to a fully recrystallized state. The average grain sizes of 1177-AC and 1177-WQ samples were measured to be ~84 μm and ~65 μm, respectively. In the 1177-AC sample, a mass of precipitates formed inside the grains and at the grain boundaries (Figure 3a). This is because the air-cooling process can provide adequate time for element diffusion and thus induce segregation/precipitation. Some of the precipitates may also come from the residual ones that were not fully dissolved during HT. In the grain interior, the diameters of the precipitates ranged from 0.8~1.2 μm and were randomly distributed (Figure 3d), while the laminar precipitates with thickness of about 0.2 μm were located at grain boundaries (Figure 3c). EDS analysis indicates that the precipitates formed after HT are mainly metallic carbides, which are Cr- and Mo-enriched $M_{23}C_6$ phase in the grain interior (Figure 3d) and laminar-structured carbides with depleted Fe, Cr, and Ni at grain boundaries (Figure 3c). Different from the 1177-AC sample, water quench can effectively suppress the formation of carbides in the 1177-WQ sample, especially the formation of lamellar-structured carbides at grain boundaries, resulting in a relatively uniform structure with some annealing twins (Figure 3b). Figure 3e shows the crystal structure of L-PBFed and HTed specimens, indicating that the specimens kept a face-centered cubic (FCC) matrix before and after HT.

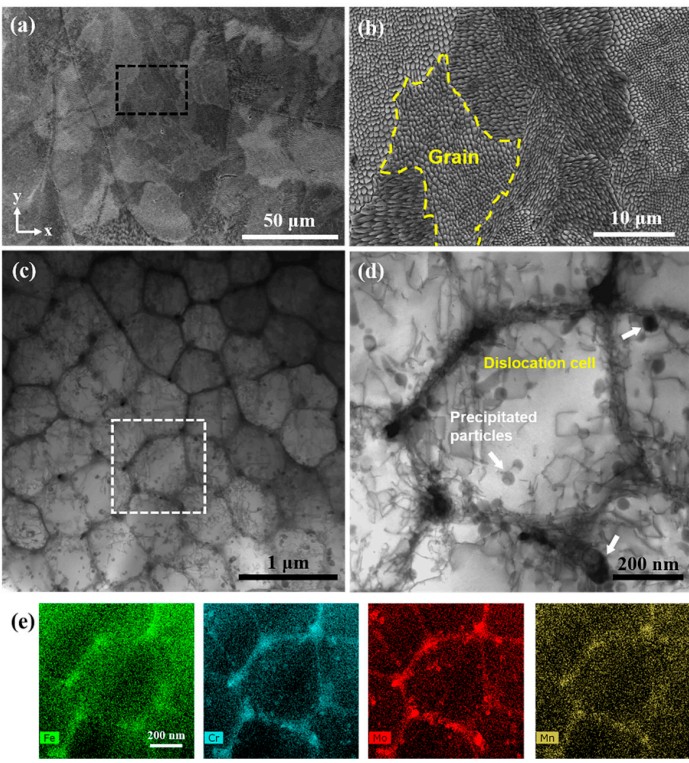

**Figure 2.** Microstructures of the as-fabricated GH3536 superalloy. (**a**,**b**) SEM images show the typical microstructure perpendicular to the building direction of L-PBF. (**c**) Bright-field STEM image shows the structure of dislocation cells with some precipitates. (**d**) Zoomed-in area marked by the white rectangle in (**c**). (**e**) Element-distribution maps of the selected area in (**c**), demonstrating the element segregation at the walls of dislocation cells.

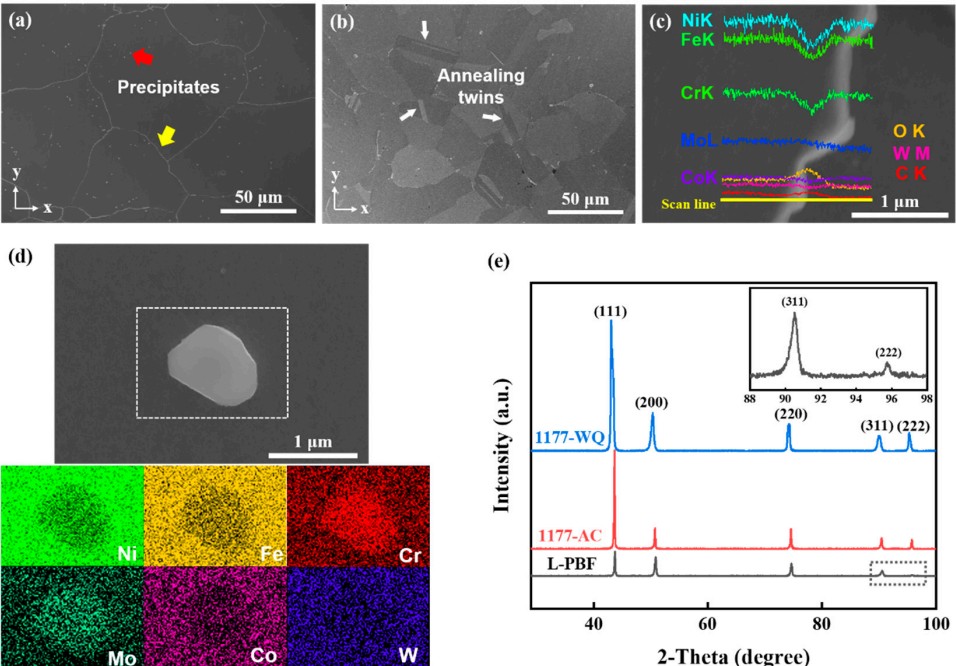

**Figure 3.** Microstructures of the HTed samples. (**a**,**b**) Microstructures of 1177-AC and 1177-WQ samples predencular to building direction, respectively. (**c**) The laminar precipitate of 1177-AC with a thickness of ~0.2 μm at the grain boundary and the EDS line-scan patterns (marked by yellow line). (**d**) EDS mapping of precipitates in the 1177-AC interior which are randomly distributed and ranging from 0.8~1.2 μm. (**e**) The XRD patterns of L-PBFed, 1177-AC, and 1177-WQ samples.

### 3.2. Mechanical Behaviors

Tensile tests were conducted at room temperature to reveal the effect of HT on the mechanical properties of L-PBFed samples. Figure 4a,b show the engineering and true stress–strain curves of as-fabricated and HTed samples, respectively. Table 2 summarizes the mechanical properties of different samples. Clearly, even with high residual stress, the as-printed samples typically have higher yield strength (667 MPa) and ultimate tensile strength (835 MPa), but a relatively low elongation as compared with that of HTed samples. After HT, the samples show decreased tensile strengths, especially the yield strength, but an enhanced elongation from 40% to 68%. The high yield strength of the L-PBFed sample mainly originates from the fine dendrites and dislocation structures according to the Hall–Patch effect [24,40]. When HTed at solution temperature, the residual stress was released and the typical characters of L-PBFed alloy disappeared by forming a homogenized microstructure with equiaxed grains [31]. The grown grains, eliminated dendrites, and relaxed dislocations could result in lower strength but much higher elongation, while the released residual stress and recrystallized grains could contribute to a higher strain-hardening ability. In the true stress–strain curve, the ultimate tensile strengths of HTed samples are even higher than those of as-printed samples due to the significant strain hardening. However, the strain-hardening rate of 1177-AC maintains a platform and is far beyond other specimens with the increase in true strain (Figure 4c). We hypothesize that due to the pinning effect of precipitates on dislocation motion, the formation of carbide precipitates on grain boundaries results in dislocation pile-up and contributes to the strain hardening. In other words, the precipitates at grain boundaries delay the necking, thus leading to a higher strain-hardening rate. However, due to the deformation incompatibility of brittle carbides, the strengthening effect of precipitates on grain boundaries should be limited. In the 1177-WQ sample, the recrystallized grains with lower precipitate density enable a more uniform elongation but lower strain-hardening capability.

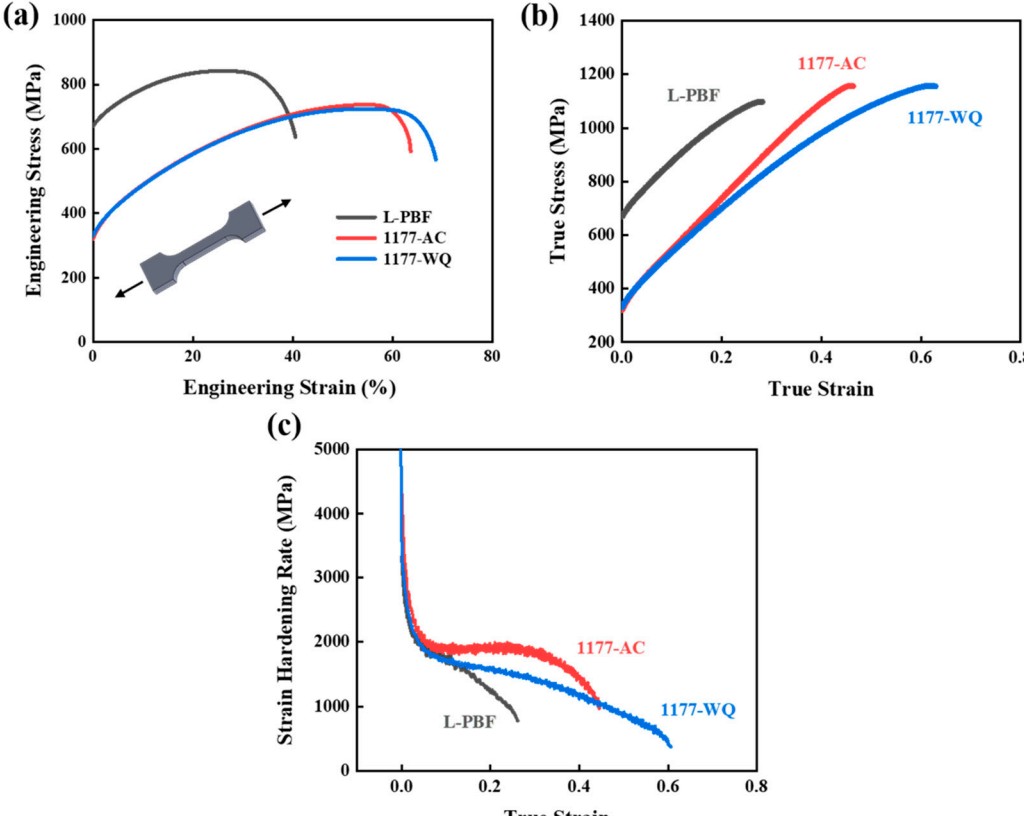

**Figure 4.** Mechanical properties of the L-PBFed and HTed samples. (**a**,**b**) Engineering and true stress–strain curves of different samples. (**c**) Strain-hardening rate of different samples.

**Table 2.** Mechanical properties of GH3536 superalloy at room temperature.

| Samples | Yield Strength (MPa) | Ultimate Tensile Strength (MPa) | Elongation (%) |
|---------|----------------------|----------------------------------|-----------------|
| L-PBF | $667 \pm 3$ | $835 \pm 6$ | $40.21 \pm 0.74$ |
| 1177-AC | $323 \pm 2$ | $731 \pm 5$ | $63.01 \pm 0.62$ |
| 1177-WQ | $334 \pm 3$ | $721 \pm 3$ | $68.70 \pm 0.04$ |

Figure 5 shows the typical fracture morphologies of different samples. The fracture surface of as-printed L-PBF samples contains several large pores at the center (Figure 5a), which should be intrinsic printing defects. Obviously, even very few of these defects could result in the generation of large pores and cracks during the tensile test. In the HTed samples, both the size and density of pores on their fracture surface are much smaller than those of the L-PBFed sample (Figure 5d,g), while the symbol of plastic deformation—dimples becomes larger, suggesting that HT can improve the ductility of the as-printed samples. Although the mechanical properties of 1177-AC and 1177-WQ samples are similar, their fracture morphologies are much different. As shown in Figure 5d, the fracture surface of the 1177-AC sample contains numerous cracks, which propagate toward the interior of the materials (Figure 5e). These cracks mainly derive from the pores and propagate zigzags, which are probably induced by the laminar-structured carbides on grain boundaries. After air cooling, the carbides precipitate out in the grain and form a continuous layer along the grain boundary. Though $M_{23}C_6$ in the grain interior can provide sites for the nucleation of dimples (Figure 5f) and increase tensile instability, the fracture of 1177-AC is mainly determined by precipitates at grain boundaries. These brittle carbide layers can act as a barrier shell to block the dislocation motion, but will induce grain-boundary fracture as the strain accumulates. With these carbides at grain boundaries, cracks possibly initiate from the inherent pores induced by the L-PBF, and then develop gradually along the brittle grain boundary. This hypothesis is confirmed by the observation of the polished fracture surface of the 1177-AC sample (Figure 6). Apparently, several transgranular cracks nucleate from or nearby a pore and then develop along the grain boundary (Figure 6a). Note that some cracks may not directly connect to the pores, but can be induced by the large deformation incompatibility at the grain boundary near a pore due to the high stress concentration. Although the crack propagation can consume some deformation energy and thus contribute to plastic deformation, the brittle grain boundary may be harmful for cyclic deformation, which thus should be avoided. In the 1177-WQ sample, elimination of the brittle carbide layer on grain boundaries largely suppresses the cracking behavior on the fracture surface, resulting in transgranular fracture (Figure 5g,h). It is worth noting that there was an agglomeration of pores in the subsurface, and the sizes of pores are larger than dimples (Figure 5h), indicating that HT could not fully eliminate the inherent pores. Few carbides are observed in the dimples of 1177-WQ (Figure 5i), contributing to the better ductility than the 1177-AC sample.

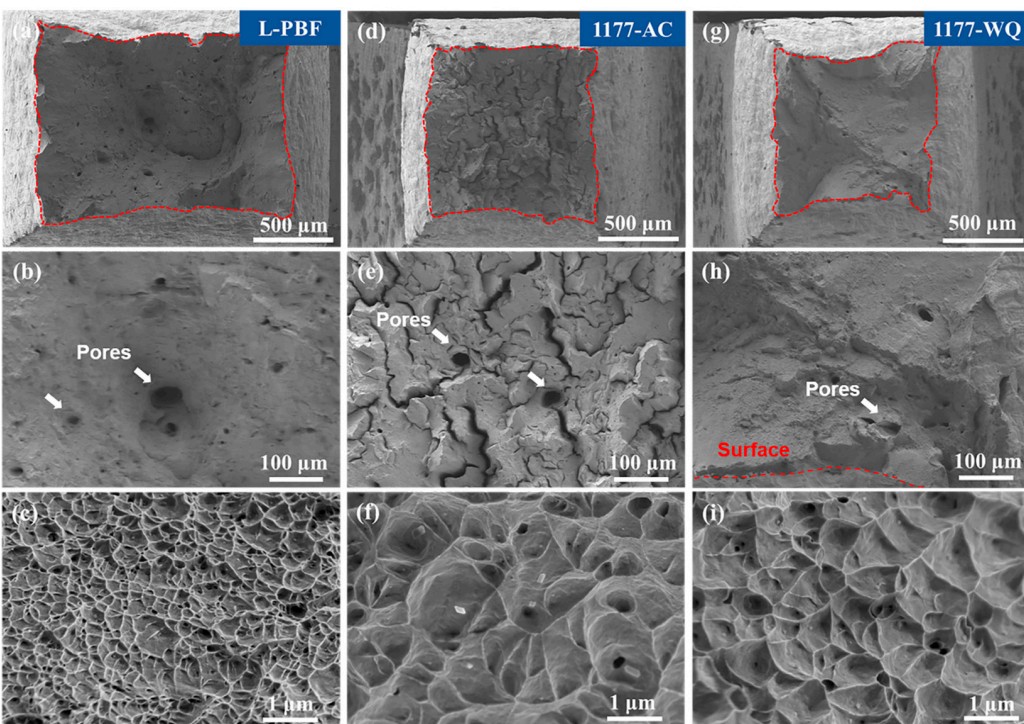

**Figure 5.** Fracture morphologies of different samples. (**a**–**c**) In L-PBF condition, (**d**–**f**) in 1177-AC condition, and (**g**–**i**) in 1177-WQ condition. The white arrows in (**b**,**e**,**h**) indicate the probable inherent pores in fracture surface. The red dotted box highlights the boundary of fracture surface.

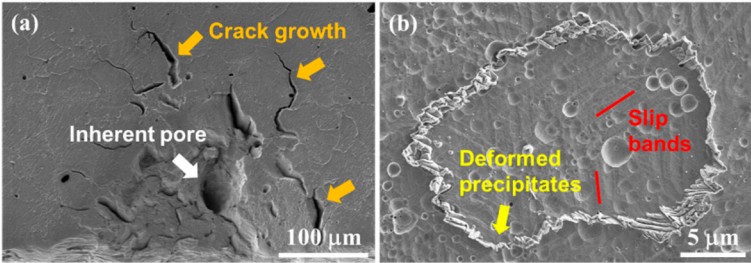

**Figure 6.** The polished fracture surface of the 1177-AC specimen. (**a**) The crack near an inherent pore. (**b**) The precipitated phases on the grain boundaries can act as a barrier for dislocation motion.

### 3.3. Structure of Deformed Samples

The high strength of the L-PBFed sample mainly originates from fine microstructure with dissatisfying low ductility limited by pores and other 3D-printing defects. In 1177-AC and 1177-WQ samples, the coarsened grains and reduced dislocation density weaken the influence of additive manufacturing defects on the mechanical properties. The homogenization and coarsened grains result in lower yield strength but higher ductility. To further understand the mechanical properties, the deformation structure of different samples was investigated. In the as-printed L-PBF sample, because of the high density of tangled dislocations, it is difficult to characterize the dislocation structure of deformed sample. In the HTed samples, the dislocation cell structure was largely destroyed, while some entangled dislocations and residual precipitates remain (Figure 7a,e), which are mainly 60° a/2 [110] full dislocations (1/6 [112] Shockley partial dislocations with stacking fault were also observed, Figure 7b,c). The previous study [41] revealed that the stacking-fault energy of GH3536 superalloy is relatively low (comparable with that of medium-/high-entropy alloys), such that full dislocations are prone to transform to extended dislocations at the early stage of plastic deformation. Since HT cannot fully eliminate the high-density dislocations produced by L-PBF, the structure of extended dislocations thus can be observed

in the HTed sample. The dislocation density of the 1177-AC sample is higher than that of the 1177-WQ sample, in which tangled dislocations interweave to form dislocation braids. This might be because a large number of solid-solution elements were precipitated from the grain, and the laminar precipitates at grain boundaries may induce large internal stress interior of the grains. Some annealing twins were observed in both 1177-AC and 1177-WQ samples (Figure 7b), which can contribute to the strain hardening via dislocation–twin interaction [42,43]. Annealing twins divided the grains into smaller subspace, contributing to the Hall–Petch-type hardening. After tensile deformation, numbers of dislocations and slip bands were generated inside the 1177-AC specimen (Figure 7d). This could be ascribed to the contribution of carbides in the grain boundary that effectively hindered the dislocation motion and resulted in a dislocation pile-up inside the grain, contributing to the enhanced strain-hardening rate during the middle stage of deformation of air-cooling samples. In the 1177-WQ sample, dislocation multiplication, pile-up, and entanglement occurred abundantly in the grain interior, near the annealing twins (Figure 7e) and at the grain boundaries. The pre-existing annealing twins decreased the mean free path of dislocations during deformation, enhancing the work hardening ability (Figure 7f). It is also worth noting that precipitates cannot be fully eliminated via the HT at 1177 °C even with the water quench process (indicated by yellow arrows in Figure 7a,e). These particles can hinder the dislocation motion and contribute to the strengthening of HTed samples, but have not been fully investigated at present.

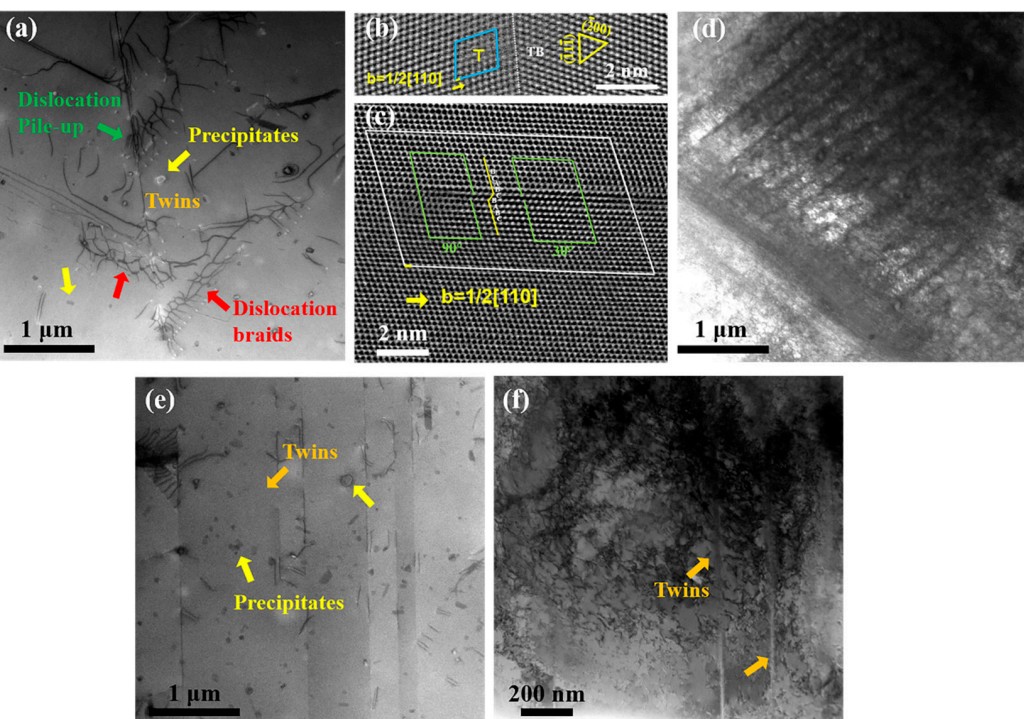

**Figure 7.** The STEM observation of the HTed samples. The 1177-AC before (**a**–**c**) and after (**d**) the tensile deformation. The 1177-WQ before (**e**,**f**) after the tensile deformation.

## 4. Discussion

### 4.1. Deformation of L-PBFed Sample

The microstructure of L-PBFed materials is mainly determined by the processing parameters and solidification conditions. Typically, the dislocations of L-PBFed components originate from the thermal expansion and contraction during solidification, and dislocation-cell structures are formed along the boundaries of the columnar dendrites, which maintain the same orientation as dendrites [38]. Due to the thermal cycling effect of laser-additive manufacturing, supersaturated elements are prone to precipitate from the alloy matrix and result in local segregation [39]. Hence, segregations in dislocation cells as well as their

boundaries can be understood. With the dense cellular dislocation structure, the yield strength of L-PBFed materials was often described by the conventional Hall–Petch effect. The grain size *d* can be determined by the diameter of the dislocation cell to fully depict the high strength of the L-PBFed sample [24,40]. However, the temperature gradient between the melting-pool boundary and its upper surface along the building direction often causes insufficient melting in the metal part [17,44,45]. These areas can significantly decrease the ductility of L-PBFed specimens, due to the deformation-induced stress concentration and high possibility of inherent pores at the melting-pool boundaries, as demonstrated by our experimental observations in Figures 4a and 5a,b.

### 4.2. Deformation of HTed Sample

HT at solution temperature can dramatically change the microstructural characteristics of the L-PBFed specimen (Figure 3a,b). Coarsened grain and homogenized microstructure after HT were favorable to enhance the tensile stability and ductility of L-PBFed samples. However, dissolved dendrites and reduced dislocation density and cell structures would decrease the tensile strength of HTed samples (Figure 4a). For HTed samples, the grain size of 1177-AC (~84 μm) is slightly larger than 1177-WQ (~65 μm). However, as shown in Figure 4, 1177-AC possesses a slightly lower yield strength, higher strain-hardening rate, and lower elongation than 1177-WQ, indicating that the carbide precipitates at grain boundaries introduced by air cooling significantly changed the mechanical behaviors of the HTed sample and decreased the tensile stability, especially the ductility of 1177-AC. Furthermore, the XRD patterns in Figure 3e show that 1177-AC possesses a lower concentration of solid-solution elements than 1177-WQ (with the average lattice parameters of L-PBF, 1177-AC, 1177-WQ about 3.5915 Å, 3.5951 Å, and 3.6190 Å, respectively), demonstrating that lots of solid-solution elements (Cr and Mo) precipitated during the air-cooling process and contributed to a lower tensile strength. However, the ultimate tensile strength of 1177-AC is slightly higher than 1177-WQ, although 1177-AC possessed larger grain size and fewer solid-solution-strengthening elements in matrix. These results further indicate that the carbide precipitates at grain boundaries can contribute to the continuing strain hardening of 1177-AC.

For the solid-solution alloys, the Taylor hardening law [41,46] was used to describe the interactions of dislocations with increasing strain rates:

$$\sigma_T = \alpha G b \sqrt{\rho} \tag{1}$$

where $\alpha$ is constant, *G* is the shear modulus, *b* is the Burgers vector, and $\rho$ is the dislocation density. The strain-hardening behaviors of 1177-AC and 1177-WQ samples are discussed according to Equation (1). At the initial stage of plastic deformation, dislocations are activated and then rapidly proliferated combining with their pileups at grain boundaries, which in turn further hinder the motion of other dislocations. As the strain increases, the density of dislocations continuously increases. As shown in Figure 7f, dislocations in the 1177-WQ sample form a complex structure through dislocation–dislocation and dislocation–twin interactions. In the 1177-AC sample, as dislocations slip to grain boundaries, the shell-like carbides segregated at grain boundaries can greatly hinder dislocation motion, by which, as the strain accumulates, high deformation incompatibility occurs, resulting in cracks along grain boundaries (Figure 5e). This deformation process is illustrated by the observation of high-density slip bands in the interior of the grains and the transgranular cracks in Figure 6a.

Figure 8 shows the enhanced ductility and acceptable ultimate tensile strength of HTed GH3536 superalloy compared with Hastalloy X in other literatures, including L-PBFed/HTed samples and wrought samples. However, numerous built-in pores were still generated during plastic deformation and can be seen on the fracture surface of HTed samples (Figure 5d,g). This means that HTed samples cannot fully eliminate the inherent defects induced by L-PBF, which can impact the crack nucleation and fracture of HTed samples. Hot isostatic pressing (HIP) is often conducted to reduce the impact of these

L-PBF-induced defects, during which carbides may grow during the thermal exposure of HIP and influence the ductility of materials [30].

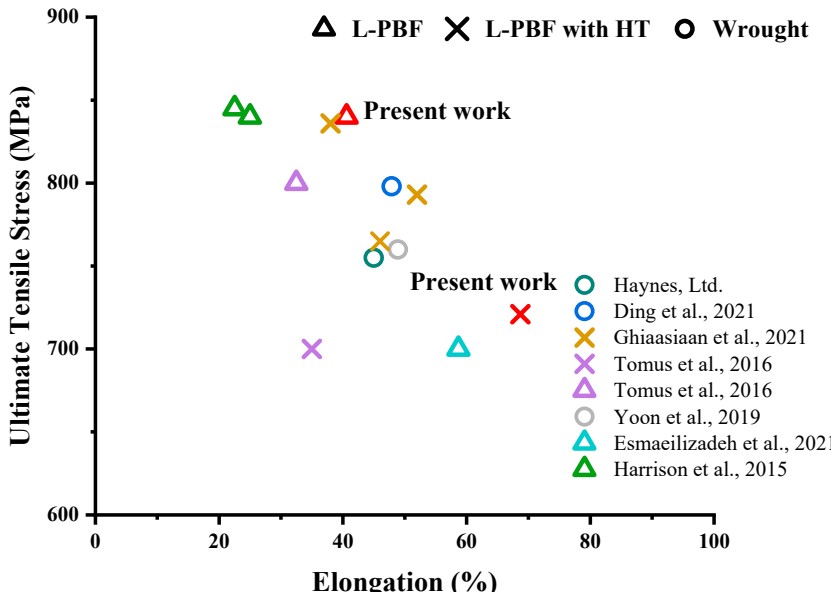

**Figure 8.** The comparison of ultimate tensile strength and ductility of GH3536 superalloy from our work and literatures [18,24,30,34,41,47,48].

## 5. Conclusions

We fabricated the GH3536 alloys via L-PBF methods and studied their microstructure and mechanical properties under different conditions. Post-heat-treatments reveal that different cooling processes result in totally different microstructure and fracture morphology in HTed GH3536 superalloy. The main results can be concluded below:

(a) A near full-dense specimen containing a high density of fine dendrites and cellular dislocations was fabricated by L-PBF, which showed high yield strength and ultimate tensile strength, but relative lower ductility.

(b) The longer homogenization time and air-cooling process lead to coarse grains but laminar precipitates at grain boundaries. The carbide precipitates may act as stable barriers that hinder the dislocation motion for strength, resulting in a higher strain-hardening rate but lower ductility.

(c) Constraining the formation of precipitates in grain boundaries by water quench is necessary for HTed samples, which contribute to an enhanced ductility and acceptable strength compared with wrought samples.

**Author Contributions:** Conceptualization, H.N., Q.Z. and J.W.; methodology, H.N. and J.W.; validation, K.Z. and J.W.; formal analysis, H.N. and J.W.; investigation, H.N. and Q.Z.; resources, Q.Z., K.Z. and J.W.; writing—original draft preparation, H.N.; writing—review and editing, H.N., Y.C. and J.W.; visualization, H.N.; project administration, J.W.; funding acquisition, J.W. All authors have read and agreed to the published version of the manuscript.

**Funding:** The authors gratefully acknowledge the support of the Aero Engine Corporation of China (Grant HFZL2019CXY001). J.W. acknowledges the support of the National Natural Science Foundation of China (Grant 52071284).

**Institutional Review Board Statement:** Not applicable.

**Informed Consent Statement:** Not applicable.

**Data Availability Statement:** Not applicable.

**Conflicts of Interest:** The authors declare no conflict of interest.

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
