# Peer review of "Mechanical Properties and Fracture Behavior of Laser Powder-Bed-Fused GH3536 Superalloy"

_metals, doi:10.3390/met12071165_

Round 1

Reviewer 1 Report

Dear authors,

I have read your manuscript with attention and interests. 

My comments which could be helpful to improve your paper were listed below:

1. Page 1, line 22 " resulting in low strength" I suggest mitigate that sentence and to write lower instead low

2. Keywords, as a rule it should not be a words repeated form the title

3. Page 1, Line 45, sentence "Among different metallic materials, nickel-based    superalloys are the most widely 45 used materials as hot parts of gas turbine"

to support this sentence, I suggest quoting two works

    1) DOI: 10.3390/ma14112745. 

     2) DOI: 10.1007/s11666-022-01400-5 4. 

4. Page 2, line 74, is "power", should be powder

5. Page 2, line 80, air-cooling process should be described more precisely because the repeatability of the experiment is essential in scientific reports.

6. Page 4, line 129, "to be ~84 129 μm and ~65 μm" Measurement according to what standard?

Conclusion should be presented in points

Regards

Reviewer 2 Report

The paper provides useful information about mechanical properties and fracture behavior of an additive manufactured superalloy. However, it should be improved for the following points:

The testing procedures, in particular tensile testing, should be referred to applicable standards. Moreover, it is unclear whether the geometry of the selected very short gauge length, dog-bone specimens with a square cross-section is conforming to any standard.

In Table 2, the properties of the L-BDF series should also be reported. It should also be commented how the properties compare with the ones of the corresponding wrought superalloy.

Finally, the English should be improved (remain existed, gridding, powers for powders etc.)

Reviewer 3 Report

The manuscript concerns the study of the microstructure and heat treatments of an alloy similar to the Hastelloy X processed by LPBF process.

The manuscript is quite interesting but several points need modifications or clarifications.

Introduction:

1    1)      Since the alloy is similar to Hastelloy X, the authors should clearly highlight the novelty of their study with respect to the literature on Hastelloy X. In the current state, the innovation is not clear.

2   2) Please provide examples to explain when is important to increase the ductility for industrial applications. Moreover, I believe that the increment of grain size for superalloys is also related to increasing their corrosion and oxidation resistance in harsh environments.

Materials and methods

3  3)      How was selected the reported process parameters?

The speed is mm/min or mm/s. Considering the speed used in other works on Hastelloy X processed by LPBF typically the speed is reported in mm/s.

4    4)      Please clarify if a pre-heating temperature was used during the process since EOS M290 allows to increase the temperature of the building platform.

5    5)      For the intercept method, how many images were considered? How many grains were analyzed?

6    6)      Please provide the parameters used for the XRD analysis.

7     7)      Please provide more details on the polishing steps.

8      8)      How was determined the chemical composition reported in Table 1?

9      9)      How many tensile specimens were tested?

Results

1    10)   I would recommend adding the SEM images of unetched samples at low magnification to show the high densification declared by the authors.

1 11)   For Figure 2: the authors should provide the EDS on the grain boundaries both for the heat treated condition after FC and WQ.

1   12)   Provide more details on the dimensions of the carbides and their position.

1    13)   It seems that the WQ condition presented annealing twin while the AC version did not present annealing twin.

114)   For Table 2 please provide also the standard deviation.

Discussion

115)   The obtained results should be compared with the available literature on Hastelloy X alloy.

Conclusions:

116)   Please re-write the conclusions in order to clearly shows the results obtained in the work.

Reviewer 4 Report

See the attached file, please. 

Round 2

Reviewer 3 Report

The authors improved the quality of the manuscript.

Please provide that the chemical composition in table 1 is obtained by EDS analysis. Moreover, I do not think that C and B can be determined by EDS with similar precision. I believe that only the main elements were determined by EDS. Please correct it in the text.

In material and methods, the authors should provide how many images were analyzed for the determination of the grain sizes.

The mechanical properties of the material should be compared to the mechanical properties of the LPBFed Hastelloy X reported in the literature.

Reviewer 4 Report

The authors partly addressed the open question raised by me. For several comments, the authors claimed to have revised the text and added the relevant discussion and references; however, I could not find any inputs in the text. As the review process is voluntary and time-consuming, I urge the authors to carefully read the comments and make sure to address them in the revised manuscript. Please consider the following comments and open questions. Please refer to the first round of revision comments for original comments and questions.

Introduction

2- The paper is still missing a brief discussion on the microstructure and mechanical properties of GH3536 alloys fabricated by conventional approaches.

Material and Methods

3-  Mention the range of the parameters tested and the optimization method.

5- I can not see the tensile testing machine type. Please add it.

6- You claimed to add details to the revised manuscript but did not. Discuss briefly in the text to address the original question.

Results

7- A discussion is required here to justify the crack-free builds in your work compared to the earlier studies.

9- A citation to the relevant reference is still missing.

10- I can not find any discussion in the revised paper.

12- No new discussion and citations could be found in the revised paper.

13- No new discussion is found.

14- No details were provided in the revised paper.

15- The authors' response is not acceptable. Figure 1 shows minor pores in the builds. The pores in the fracture surfaces might form during the tensile testing.

16- The authors' response is unacceptable as they performed independent research. If they want to mention the improved density after HT, they should investigate the porosity change before and after the HT.

17- I can not see a discussion on the possible effect of residual stress reduction after HT on the strength in the revised paper.

19- I can not see any discussion on the dislocation-precipitates interactions and pinning effect of precipitates on dislocation motions in the revised paper.

New comments:

In Fig. 1, it is required to indicate the building and scanning directions.

Table 1 lists the nominal composition of the GH3536 superalloy powder. Correct it in the caption. 

Round 3

Reviewer 4 Report

The authors sufficiently addressed the questions. I recommend the article to publish.